# Efficacy of a Six-Week Dispersed Wingate-Cycle Training Protocol on Peak Aerobic Power, Leg Strength, Insulin Sensitivity, Blood Lipids and Quality of Life in Healthy Adults

**DOI:** 10.3390/ijerph17134860

**Published:** 2020-07-06

**Authors:** Chun Hou Wun, Mandy Jiajia Zhang, Boon Hor Ho, Kenneth McGeough, Frankie Tan, Abdul Rashid Aziz

**Affiliations:** 1Department of Physiology, Yong Loo Lin School of Medicine, National University of Singapore, Singapore 117597, Singapore; chunhou.wun@gmail.com (C.H.W.); frankie_tan@sport.gov.sg (F.T.); 2Changi Sports Medicine Centre, Changi General Hospital, Singapore 529889, Singapore; Mandy_Zhang@cgh.com.sg (M.J.Z.); Boon_Hor_ho@cgh.com.sg (B.H.H.); 3ActiveSG, Active Health Division, Sport Singapore, Singapore 397630, Singapore; kenneth_mcGeough@sport.gov.sg; 4Sport Science and Sport Medicine, Singapore Sport Institute, Sport Singapore, Singapore 397630, Singapore

**Keywords:** aerobic, anaerobic, high-intensity interval training, sprint interval training, cardiometabolic markers

## Abstract

*Background*: The aim of this study was to evaluate the efficacy of a six-week dispersed Wingate Anaerobic test (WAnT) cycle exercise training protocol on peak aerobic power (VO_2peak_), isokinetic leg strength, insulin sensitivity, lipid profile and quality of life, in healthy adults. *Methods*: We conducted a match-controlled cohort trial and participants were assigned to either the training (intervention, INT, *N* = 16) or non-training (control, CON, *N* = 17) group. INT performed 30-s WAnT bouts three times a day in the morning, afternoon and evening with each bout separated by ~4 h of rest, performed for 3 days a week for 6 weeks. Criterion measures of peak oxygen uptake (VO_2peak_), leg strength, insulin markers such as homeostatic model assessment (HOMA) and quantitative insulin-sensitivity check index (QUICKI), blood lipids profile and health-related quality of life (HRQL) survey were assessed before and after 6 weeks in both groups. *Results*: Absolute VO_2peak_ increased by 8.3 ± 7.0% (*p* < 0.001) after INT vs. 0.9 ± 6.1% in CON (*p* = 0.41) group. Maximal voluntary contraction at 30°·s^−1^ of the dominant lower-limb flexors in INT increased significantly post-training (*p* = 0.03). There were no changes in the INT individuals’ other cardiorespiratory markers, HOMA, QUICKI, blood lipids, and HRQL measures (all *p* > 0.05) between pre- and post-training; but importantly, no differences were observed between INT and CON groups (all *p* > 0.05). *Conclusions*: The results indicate that 6 weeks of dispersed sprint cycle training increased cardiorespiratory fitness and dynamic leg strength but had minimal impact on insulin sensitivity, blood lipids and quality of life in the exercising individuals.

## 1. Introduction

Exercise and a well-balanced diet are the cornerstone of good health. Engaging in regular aerobic-type of exercise has proven to elicit several beneficial adaptations including increased cardiorespiratory fitness, and enhanced glycemic control [1]. In line with these benefits, the World Health Organization (WHO) recommends that adults engage in a minimum 150 min per week of moderate-intensity or 75 min per week of vigorous-intensity continuous aerobic physical activity to achieve meaningful improvements in cardiorespiratory fitness and metabolic health [2]. Unfortunately, less than 50% of the population cohort studied have met these physical activity recommendations, often citing ‘lack of time’ as a common barrier to exercise [3,4]. In this regard, high-intensity interval training (HIIT) has been proposed as a time-efficient intervention that can bring about similar benefits as compared to the typically much longer duration of moderate-intensity continuous training (MICT) [5,6].

HIIT is described as an intermittent period of short intense exercise interspersed by recovery period [6]. Two main forms of HIIT exist, known as ‘high-intensity training’ (HIT) and ‘sprint interval training’ (SIT), respectively. HIT incorporates bouts of close to maximum intensity exercise performed between one to several minutes at between >80% peak oxygen uptake (or VO_2peak_) to <100% VO_2peak_, with several minutes of low-intensity recovery between bouts. SIT on the other hand, consists of bouts of supramaximal, i.e., ‘all-out’ effort at >100% VO_2peak_ with passive rest or light active recovery between each sprint bout [6]. 

The scientific literature has shown the effectiveness of SIT in enhancing cardiorespiratory fitness [7]. The mechanisms underlying skeletal muscle metabolic adaptations to SIT however, have not fully been elucidated [8,9]. Studies have shown that SIT increases mitochondrial capacity through activation of peroxisome-proliferator activated receptor coactivator (PGC)-1α. PGC-1α is also known as the ‘master switch’ of mitochondrial biogenesis in muscle cells which participates in the regulation of carbohydrate and glucose uptake, oxidative capacity, anti-oxidant defense and anti-inflammatory pathways [6,9]. The SIT protocol of low-volume ‘all-out’ sprint bouts has the potential to increase PGC-1α messenger-ribonucleic acid (mRNA) by multiple folds when measured post exercise [8], which is comparable to the effects observed after a bout of continuous moderate-intensity endurance-type exercise [10,11]. Indeed, as little as six exercise sessions of 4 to 7 bouts of 30-s ‘all-out’ sprints with 4 min recovery performed over two weeks, positively impacted the muscle’s maximal capacity to use oxygen, or otherwise known as skeletal muscle oxidative capacity [12]. 

A typical example of the traditional SIT exercise protocol employs the Wingate anaerobic cycle test (WAnT) which consists of two to four 30-s bouts of ‘all-out’ or supramaximal efforts (>100% VO_2peak_) performed consecutively and interspersed with only between ~2 to 4 min of active or passive recovery periods [12,13]. The traditional SIT exercise protocol is likened to a ‘clustered’ protocol. This ‘clustered’ WAnT protocol is known to be physically strenuous and requires high levels of self-motivation to complete successfully [14,15,16]. Thus, for clustered WAnT training protocol to be more appealing, we proposed to increase the rest periods between each of the sprint bout, in what we termed as the ‘dispersed’ WAnT protocol. In this dispersed protocol, a single 30-s WAnT bout is performed three times a day, but with ~4 h of rest between each bout. Using this exercise protocol, for the same volume and/or number of sprint bouts per day (e.g., 3 × 30-s cycle bouts), the dispersed protocol requires less than half the amount of time to complete as compared to the clustered protocol due to the elimination of recovery between the bouts (see Table 1). Thus, the dispersed protocol is deemed to be more time-efficient. Previous research has indicated that individuals can accrue similar health benefits from exercising in a single bout or accumulating activity from shorter bouts throughout the day [17]. With the concept of ‘accumulated’ benefits of exercise, the dispersed WAnT protocol may be a potential exercise strategy for individuals who are unwilling or unable to spent time in prolonged physical activity. Indeed, two recent training studies that employed the dispersed WAnT protocol observed improvements in VO_2peak_. The first was a pilot study conducted by our research group on sedentary females, consisting of 3 × 30-s WAnT sprint bouts with 4 h of rest between bouts, 3 days per week for 8 weeks. We showed an average of ~14% improvement in VO_2peak_ [18]. Another study that employed a similar protocol also reported that 3 × 20-s WAnT sprint bouts with 1 to 4 h rest duration performed for 3 days per week for a total of 6 weeks, led to ~4% improvement in VO_2peak,_ among inactive but moderately fit college-aged individuals [19]. These two studies showed that WAnT bouts performed in a dispersed fashioned could result in modest improvements in cardiorespiratory fitness in trained and untrained individuals. However, several limitations were noted in these two studies. These include the lack of non-training control group to quantify the magnitude of training-induced improvements, and the effects of the dispersed WAnT protocol on other important health markers such as insulin sensitivity, lipid profile, functional strength, and perceived quality of life were not assessed. 

Therefore, our aim in the present study was to address these limitations and provide additional evidence of the efficacy of a dispersed WAnT cycle training protocol. Based on the findings of previous studies [18,19], the working hypothesis of the present study was that a dispersed WAnT exercise training protocol is effective in improving cardiovascular fitness, leg strength, insulin sensitivity and lipid profile of healthy individuals.

## 2. Methods

### 2.1. Participants

A total of thirty-four healthy male and female participants, aged 21–45 y, were recruited from ~320 staff working for an organization whom had convenient access to the laboratory facilities where all testing and training sessions were conducted. A validated pre-screening physical activity tool was administered to all volunteers, followed by a screening process performed by a sport physician. This is to ensure that participants were free from risk factors associated with metabolic, cardiovascular or pulmonary disease, and have had no pre-existing musculoskeletal conditions that would limit their physical function. Participants were also informed that lifestyle changes may potentially affect the outcome of the study, and hence, were instructed to maintain their routine diet and daily activities. The experimental procedures and potential risks were fully explained to all participants prior to obtaining their written-informed consent. The study was approved by the institutional review board and conformed to the Declaration of Helsinki. 

A match-controlled cohort trial was carried out and volunteers were assigned to either the training/intervention (INT) or non-training/control (CON) group. Several blocks of recruitment exercise took place over a 16-week period. Approximately 4 to 8 participants were recruited during each block, and they were tested for their pre-training or baseline VO_2peak_ before being assigned to either the INT or CON groups based on their VO_2peak,_ age and body mass index (BMI), to ensure similar baseline characteristics for both groups.

### 2.2. Pre- and Post-Experimental Procedures

All participants arrived in the laboratory in the morning after at least 8 h of overnight fasting. They were advised to consume 500 mL of water in the morning of the test. They were instructed to abstain from strenuous exercise 2–3 days before the test and avoid alcohol and smoking the evening before the test. Baseline measurements for all participants consisted of the assessment of basic anthropometric measures, resting blood pressure (BP), resting heart rate (HR), body fat, VO_2peak_, blood lipids, fasting blood glucose and fasting insulin, as well as a survey on their quality of life. After the completion of 6 weeks of training, post-training tests similar to baseline measures were obtained to determine the efficacy of the training intervention. The post-training measures were conducted between 3 to 7 d after the final training session. 

Anthropometric measures, resting heart rate and resting blood pressure: Stature was measured in centimeters without shoes. Body mass and body fat percentage were determined using bioimpedance technique (Inbody770, Chungcheong, Korea) following the manufacturer’s standard procedures. Resting BP and HR were assessed in a seated position, with the individuals seated upright on a chair with a back support and arm positioned at chest level for a minimum of 3 min before any measurement was taken. BP and HR measures were manually taken by the same trained physician throughout the study. Two blood pressure measurements were taken at each time point, with a 2-min rest between measures, and these were averaged to reduce variability.

### 2.3. Peak Oxygen Uptake

All participants performed a progressive exercise ramp test on a cycle ergometer (Lode BV, Excalibur Sport, Groningen, The Netherlands) to determine their peak oxygen uptake or VO_2peak_. Participants completed a standardized 3 min warm-up between 40 to 60 W. For the VO_2peak_ test, the cycle resistance was increased by 1 W every 3 s until volitional exhaustion. Volitional exhaustion is determined as the physical limit beyond which the participant was no longer able to continue the prescribed pedal rate, i.e., between 50 revolutions per min (rpm) for women and 60 rpm for men. Subsequently, after a passive rest of ~8 min, a verification test of the initial VO_2peak_ value was undertaken. Based on the peak power attained in the initial test, the individual cycled at 50%, 70%, 105% for 60-s, followed by cycle to volitional exhaustion at 120%. The expired respiratory gases during initial and verification tests were measured using a calibrated metabolic cart (TrueOne 2400MMS, Parvomedics, East Sandy, UT, USA), and VO_2peak_ was recorded as the highest value over a 30-s period during the initial and verification tests. The coefficient of variation for VO_2peak_ measurement in our laboratory is 2% (unpublished data).

### 2.4. Leg Strength

As the criterion measure of dynamic leg strength, maximal voluntary contraction (MVC) was assessed via the maximal knee extension and flexion of the dominant leg on a isokinetic dynamometer (Biodex System 3 PRO dynamometer, Biodex Medical Systems, Shirley, NY, USA). Participants warmed-up by cycling for 8 min followed by 3 min of dynamic stretching of the hip, knee and ankle joints. Participants were then securely strapped across the upper body and the hips while seated on the dynamometer chair with the arms folded across the chest. The dynamometer lever arm was secured 2–4 cm above the ankle malleolus of the dominant leg and the participant’s knee joint was then adjusted to align with the axis of rotation of the dynamometer lever arm. The range of motion of the tested leg was then set accordingly for the knee extension/flexion movement. As part of familiarization, each participant was asked to perform 5 repetitions (rep) at 300°·s^−1^, followed by 3 rep at 30°·s^−1^ at their perceived 50–60% submaximal effort. Subsequently, the participant was then instructed to forcefully extend and flex the lower leg as fast as possible, at the tested speed of 30°·s^−1^ and 300°·s^−1^, respectively. Each individual had to perform three reps of flexion and extension interspersed with ~15-s of passive rest between each rep. MVC was defined and recorded as the highest peak torque (in Newton-meters, N·m) of the 3 rep. The dynamometer was calibrated prior to each testing session and was corrected for gravity. 

### 2.5. Lipid Profile 

Following at least 8 h of overnight fasting, venous blood sample (~5 mL) was obtained using BD vacutainer blood collection system. The blood was kept in the refrigerator at 4 °C and transported to an accredited commercial biochemistry laboratory (Quest Laboratories, Singapore) within 2 h of the blood draw. Analyses for fasting glucose, fasting insulin, plasma triglyceride, low-density lipoprotein (LDL)-cholesterol and high-density lipoprotein (HDL)-cholesterol were performed using standardized procedures. The intra-assay coefficients of variation of these markers are within 3% (Quest Laboratories, Singapore). 

### 2.6. Insulin Resistance Markers

Fasting insulin and fasting glucose were obtained from venous blood sample (~3 mL) after an overnight fast for at least 8 h. Two indirect indices, the Homeostatic model assessment (HOMA) and Quantitative insulin-sensitivity check index (QUICKI), were used to estimate insulin resistance as these measures are identified as reliable and simple indirect methods for detection of insulin sensitivity [19]. HOMA index uses the formula previously described [20]: Insulin (UM^−1^) × [Glucose (mmol·L^−1^/22.5]. The QUICKI is based on the logarithmic transformation: 1/(log insulin log glycaemia in mg·dL^−1^) [21]. The blood measures of glucose and insulin were performed under standardized procedures in an accredited biochemistry laboratory (Quest Laboratories, Singapore). The intra-assay coefficient of variation for fasting glucose and fasting insulin measures are within 3% (Quest Laboratories, Singapore).

### 2.7. Psychological Measures

Health-related quality of life (HRQL) was assessed using the validated Short Form-36 (SF-36) RAND questionnaire [22]. This self-evaluated questionnaire assesses the individual’s perceived health over the training period that includes eight sub-scales grouped into the physical component summary (PCS) and mental component summary (MCS). An algorithm generates the total score using the Orthotool kit SF-36 calculator. Each subcomponent was scored using a scale ranging between 0 and 100 (arbitrary units, au). The description of the scores are: 0–20 = ‘very bad’, 20–40 = ‘bad’, 40–60 = ‘moderate’, 60–80 = ‘good’ and ‘optimal’ = 80–100 HRQL [23], whereby a score closer to 100 indicates a higher quality of life.

### 2.8. Exercise Training Intervention

Participants allocated to INT group performed the dispersed WAnT exercise training protocol on a cycle ergometer (Wattbike Pro, Nottingham, UK). They were required to perform a single 30-s ‘all-out’ sprint bout three times a day, for 3 days per week over 6 weeks, for a total of 54 bouts during the study period. Due to the challenging nature of the SIT, the duration of each cycle sprint bout commenced with 15-s for training week 1, then increased to 20-s per bout for week 2 and finally to 30-s from week 3 to week 6. On each training day, the participant performed a total of 3 sprint bouts. Every sprint bout was preceded with a 60-s warm-up at ~40 W or 50 W resistance, followed by a single ‘all-out’ sprint effort. All exercise sessions were conducted in the laboratory situated within the participants’ working environment; the estimated walking distance from the participants’ workstations to the laboratory was between 2 to 5 min. In addition, participants were strongly encouraged to perform the cycle sprint exercise in comfortable work attire, without the need to change into sports gear (Figure 1). About 398 of the exercise sessions (~70% of total of 568 sessions of the study) were performed in the individuals’ working attire. 

It was purposefully designed that each cycle sprint bout was interspersed with at least 4 h of rest that corresponded to their work schedule. Individuals in the INT group were encouraged to perform their first WAnT cycle (i.e., the morning bout) before work (08:30 to 10:30), the second bout around lunch time (afternoon bout; 12:30 to 14:30), and the last bout before leaving the office (evening bout; 16:30 to 18:30). This schedule was to mimic the work pattern of a regular office worker. In the present study, because of work exigencies however, the duration of the rest periods varies between and within individuals. We recorded the actual rest duration between the morning and afternoon bouts, and between the afternoon and evening bouts, in the INT group throughout the study (see Table 2).

Ratings of perceived exertion (RPE, Borg’s 0–10 scale) [24] and their affect (“how enjoyable is this exercise”, using Likert scale from 1 = ‘not enjoyable at all’, 4 = ‘moderately enjoyable’, to 7 = ‘extremely enjoyable’), were administered to participants at the end of each exercise session and prior to leaving the laboratory. The value of RPE and enjoyability post-training were summed every week across the entire training period. Participants used the same cycle ergometer throughout the 6 weeks of training. All training sessions were closely supervised by the two primary researchers. Participants set their preferred cycle resistance with the assistance of the researchers during the familiarization phase with the objective of exerting maximal power output during each cycle sprint. In the instance that a participant missed a training session, a make-up session was organized on the next available date, with the aim of completing all 54 training sessions within 6 ± 1 week. Participants in the CON group were instructed to maintain their usual lifestyle habits and behavior patterns throughout the study duration. A weekly reminder was made through telephone or text message to remind them to not to engage in any unaccustomed exercise and to keep to their normal diet.

## 3. Statistical Analysis

The sample size of 16 individuals in each group was adequate to estimate the mean of VO_2peak_ scale between two groups in healthy individuals with effect size: (5 points difference with standard deviation 6.5) using a one-way analysis of variance (ANOVA) test (two-tailed, alpha = 0.05 with statistical power 0.80) including 10% attrition rate. All numeric data were presented as mean ± SD. All statistical data were completed using SPSS version 22 (IBM Corp., Armonk, NY, USA). Data were assessed by a Shapiro-Wilk test for normality, and by a Levene test for the homogeneity of variance assumption. The Greenhouse-Geisser correction was used to account for the sphericity assumption of unequal variances across groups. Baseline values of each variable were compared between INT and CON groups by a *t*-test. Changes in the dependent variables from the baseline to post 6 weeks training were compared between the groups by a mixed-design two-way ANOVA for raw data, and their normalized changes were compared between groups by a *t*-test. Two-way mixed model ANOVAs (group [INT, CON] × time [pre, post]) were performed to test the effects of training on VO_2peak_, total cholesterol, HDL, LDL, triglycerides, fasting insulin, fasting glucose, HOMA, QUICKI and SF-36. A significant training X group interaction was used to identify training-induced changes in these variables. When ANOVA showed a significant interaction (group × time) effect, Tukey’s post-hoc test was used to detect differences between means. Time course data (i.e., peak power, mean power, weekly level of enjoyability and RPE) were compared via repeated measures of ANOVA. The effect size for the difference in the magnitude of the change from pre-training to post-training between the INT and CON groups was calculated for each outcome measure using Cohen’s *d*, and 0.2, 0.5, and 0.8 were considered as a small, medium, and large effect, respectively. Statistical significance was set at *p* < 0.05. 

## 4. Results

Of the 34 participants enrolled in the study, one participant in the INT group withdrew due to prolonged febrile illness. Thus, 33 participants (INT, *N* = 16; CON, *N* = 17) completed the study. 

INT participants completed 861 of 864 WAnT bouts in total, i.e., training adherence of 99.6%, with no injuries or clinical complications sustained over the period of the study. At baseline, both INT and CON groups had similar physical and physiological characteristics, strength and metabolic-health markers (Table 3). We have also included the pre- and post- 6 weeks of intervention results of both groups as Appendix A. 

There were significant interaction effects in absolute and relative VO_2peak_ (*F_1,31_* ratio = 8.77, *p* = 0.006 and *F_1,31_* ratio = 8.85, *p* = 0.004, respectively). Comparison between pre- to post-training indicates that there was significant improvement in both the absolute and relative VO_2peak_ in the INT group (*p* = 0.004 and *p* < 0.001, respectively) but not in the CON group (*p* = 0.42 and 0.64, respectively) (Figure 2).

The magnitude of increase in VO_2peak_ was 2.7 ± 2.1 mL·kg^−1^·min^−1^ (*p* < 0.001) in the INT group and 0.32 ± 2.51 mL·kg^−1^·min^−1^ (*p* = 0.62) in the CON group; with a large effect size difference between the two groups (Figure 3).

There were no significant time and interaction effects in body fat %, resting BP and resting HR, within and between the two groups (all *F* ratio values with *p* > 0.05; Figure 4). 

For leg strength measures, there were interaction effects for MVC at 30°·s^−1^ for flexion and extension (*F_1,31_* ratio = 4.36, *p* = 0.045 and *F_1,31_* ratio = 4.42, *p* = 0.04, respectively), and for MVC at 300°·s^−1^ for flexion (*F_1,31_* ratio = 5.00, *p* = 0.03). However, post-hoc analysis indicates only MVC at 30°·s^−1^ for flexion was statistically significantly different between INT and CON groups (*p* = 0.03, with a moderate effect size; Figure 5). 

Fasting insulin, fasting glucose, insulin sensitivity markers of HOMA, QUICKI and blood lipids did not show any time nor interaction effects between the two groups (all *F* ratio values with *p* > 0.05; Figure 6). 

There was also no significant difference in cycle sprints peak (*F_1,285_* ratio = 0.12, *p* = 0.89) and mean power (*F_1,285_* ratio = 0.01, *p* = 0.99) between the morning, afternoon and evening WAnT bouts, throughout the 6 weeks (Table 4).

In relation to the quality-of-life survey, no significant group x time interactions were detected between INT and CON, at pre- and post-training. During training week 1 and week 2 when sprint bouts were shorter (15-s and 20-s, respectively), participants generally indicated a higher level of enjoyment to the exercise compared to training week 3 to 6; although they were not statistically different across the training weeks, *F_1,94_* ratio = 0.66, *p* = 0.65 (Figure 7). RPE showed a significant time effect (*F_1,94_* ratio = 3.04, *p* = 0.014), but post-hoc analysis showed no statistically significant differences between any of the training weeks, all *p* > 0.05 (Figure 7).

## 5. Discussion

The key findings of this study were that a dispersed WAnT exercise training protocol on cycle ergometer for 6 weeks enhanced cardiovascular fitness and dynamic lower-limb muscular strength in healthy adults. In the present study, 6 weeks of 30-s sprints, 3 bouts per day, separated by ~4 h rest, 3 days per week was effective in improving aerobic fitness by ~8% compared to a non-training group. 

The present study’s findings corroborated the results of earlier studies. Using a similar dispersed WAnT exercise protocol, Little and colleagues reported a ~4% increase in VO_2peak_ following 6 weeks of 20-s sprints, 3 bouts per day, separated by 1–4 h rest for 3 days a week [19]. And our own pilot study of 8 weeks of 30-s sprints, 3 bouts per day, separated by ~4 h rest, 3 days per week, reported ~14% improvement in VO_2peak_ [18]. Collectively, these three studies contribute to the growing evidence that very short duration of ‘all-out’ supramaximal sprint cycle bouts performed multiple times throughout the day, with extended rest duration between the bouts can result in modest improvements in cardiorespiratory fitness within a wide range of healthy individuals. Interestingly, despite the brief duration of exercise commitment (exercise duration ~1.0–1.5 min), the magnitude of training induced improvements in cardiovascular fitness improvement is comparable to longer, traditional SIT using a clustered protocol (exercise duration ~10.0–15.0 min) [7]. Rapid improvements in physiological adaptation with a clustered WAnT protocol have been attributed to increased cardiac output and peripheral oxygen extraction in response to increased muscle oxidative potential [25]. In a clustered WAnT exercise protocol, it has clearly been shown that individuals were not able to sustain their maximum power output throughout all of the three 30-s bouts [26]. In contrast, in the present dispersed WAnT exercise training protocol, there were no significant differences in the average peak and mean power obtained during the morning, afternoon and evening bouts across the 6 weeks (Table 4), which indicates that individuals were able to sustain their maximal power output exertion for all of the three 30-s bouts performed throughout the day, i.e., the overall exercise intensity across the entire training programme would theoretically, be higher in the dispersed compared to the clustered protocol. Perhaps, the consistency of being able to maximize the level of exercise intensity for every single WAnT bout performed throughout the 6 weeks of training is the key stimulus that allows the dispersed protocol to be as, if not more, potent than the same number of WAnT bouts performed in a clustered fashioned [27]. However, to our knowledge, no direct study has been undertaken to determine the underlying physiological mechanism(s) of the training-induced adaptations to a dispersed WAnT exercise training protocol, which implies a fertile area for future research. 

The present study’s low volume, maximal-intensity WAnT training bouts dispersed throughout the day significantly improved leg strength as indicated by the increased in MVC at 30°·s^−1^ for flexion. The findings are in sharp contrast with two previous studies that examined the impact of a clustered WAnT training protocol on leg strength. Astorino and colleagues [28] had males and females performed a clustered WAnT protocol 6–8 times over a three-week period and their results showed no significant changes in their subjects’ MVC of the knee flexor and extensor muscle torque. Similarly, Bagley et al. [29] found that 12 weeks of clustered WAnT training did not increase the knee muscular torque in their participants, although the same subjects showed increase in their lower-limb muscle mass. We were unable to provide any mechanism(s) nor reasons for the contrasting observations between these studies, but it is likely that the same reason of the ability to elicit the greatest peak power and mean power possible during each of the three daily WAnT bouts and thus an overall higher resistance throughout the entire training programme relative to the previous studies is a key difference to the stark improvement observed in the lower limb strength in the present study. Nonetheless, our results showed a modest increased in the INT group’s knee flexor strength, which may suggest that a dispersed WAnT exercise training protocol may enhance the functional working capability and physical function of an individual’s lower-limbs. This positive impact on lower-limb strength may have important implications in exercise programming for the elderly, which requires further investigation. 

WAnT bouts training performed in the clustered fashioned has been shown to accrue many positive benefits such as greater glucose uptake by skeletal muscle, increased intramuscular expression of glucose transporter type 4 (GLUT-4), and greater depletion of intramuscular glycogen that may explain the observed improvements in insulin sensitivity in some studies [6,30,31]. In the present study however, six weeks of dispersed WAnT training was not sufficient to improve insulin resistance nor various metabolic health markers when measured 3 to 7 days after the final exercise training session. It should be noted that improvement in aerobic fitness is not obligatory for training-induced improvements to cardiovascular risk factors [32] and that factor such as heterogeneity (as the case in the present study due to the use of both males and female as subjects) caused by genetic variability such as the expression of RNA, may contribute to training susceptibility [33]. Nonetheless, the current study’s findings did corroborate with those of earlier studies conducted on healthy asymptomatic adolescents. For example, Buchan and colleagues [34] reported no change in fasting glucose or insulin following 7 weeks of 4 to 6 × 30-s maximal sprints 3 times per week. Further, a published meta-regression analyses suggest that to achieve a reduction of HOMA-IR of 0.5 units, the baseline HOMA needs to be at least 3.18, indicating that insulin sensitivity can only be improved in those who are already insulin resistant [35,36]. It is possible that the minimum threshold of insulin resistant was not sufficiently met in the present study’s healthy participants. Indeed, when exercise volume was increased and sprint bouts were performed in a clustered fashioned of 3–4 times × 30-s or 7–12 times × 60-s, [37,38,39], improvements in insulin sensitivity, albeit small in magnitude, were observed. We speculate that the differing energy fluxes and substrate utilization during the clustered and dispersed WAnT exercise protocol may be the reason for the contrasting results. During the first 30-s sprint bout of the clustered WAnT exercise protocol, adenosine triphosphate (ATP) demand is primarily met via phosphocreatine breakdown and glycolysis, with oxidative metabolism contributing to about 30% of overall energy requirements [40]. In subsequent bouts, the proportional contribution from oxidative phosphorylation is progressively increased. With the partial recovery between bouts, aerobic energy contribution progressively increases in the subsequent 30-s bouts, even though the exercise is of maximal physical effort. For the dispersed protocol, each of the 30-s morning, afternoon and evening bout of WAnT exercise would predominantly be fueled via ATP and glycolysis processes, which meant a greater anaerobic contribution in total relative to that of the clustered WAnT protocol. As such, it is assumed that there would be a greater energy flux through the oxidative mitochondrial generated substrate pathways during a clustered WAnT compared to a dispersed WAnT exercise training protocol, and correspondingly, greater insulin sensitizing effect in the clustered WAnT protocol [41]. 

Research focused on blood lipid changes after exercise of maximal and supramaximal efforts have shown contrasting outcomes [42]. Some studies reported significant positive changes [43,44,45], while others [46,47] did not observe any improvement in blood lipid profile following chronic HIT. Racil and colleagues observed an improvement in blood lipids, cardiometabolic markers and blood leptin in obese adolescent females after 12 weeks of 12 × 30-s running sprints at 50% maximal aerobic speed, three times a week [44]. Similarly, Koubaa et al. noted significant improvement in blood lipids in obese children following 2 min run at 80% VO_2peak_, performed three times a week over 12 weeks [43]. On the other hand, Crouse and colleagues did not show any influence of exercise intensity on lipid profile following a progressive exercise of 15 to 60 min three times per week for 24 weeks of cycle ergometer training at intensity of either 50% and 80% VO_2max_ [46]. Ouerghi and colleagues reported an improvement in lipid profile in sedentary obese but not in the normal weight individuals following three times a week for 8 weeks of 2 × 30-s running sprint at 100–110% maximal aerobic velocity [48]. It remains plausible that lipid lowering effects following HIIT-type of exercise is more pronounced in metabolically challenged individuals than in normal weight individuals. It may also be that there is a relationship between body fat and cholesterol levels, insofar a relatively higher volume of training relative to the present study’s SIT programme, is required to elicit changes in fat mass [47]. In short, the volume of training completed as opposed to the intensity of exercise training may be the key factor to improving lipid profile, and as such, the low volume albeit supramaximal-intensity exercise protocol in the present study may not have significant impact on blood lipids.

The design of the current study’s exercise protocol provides a practical prescription that can be easily incorporated into an office worker’s daily routine, to form an integral component between work and active lifestyle. Thus, instead of having individuals to set aside time for exercise, an exercise program built into their daily routine can help to eliminate the common excuse of ‘lack of time’ barrier to exercise [15]. Participants were further encouraged to exercise in their work attire without the need to change into specific exercise or sportswear, which saved even more time and improved efficiency. This dispersed WAnT exercise training protocol reduced training time by 60% compared to a traditional clustered protocol; to a mere 4.5 min per day or 13.5 min per week (Table 1). The side effects of ‘faints, respiratory events, nausea, light-headedness, and vomiting’ that have been associated with a clustered WAnT protocol is well-known [49]. We thus were able to circumvent these negative responses by introducing a prolonged rest periods between each cycle sprint bout. It is heartening to note that the present dispersed WAnT exercise training protocol achieved >99% compliance or attendance rate and was rated as ‘fairly’ enjoyable in the study. In addition, there were no adverse clinical events over the six weeks of training. These suggest that a dispersed WAnT exercise training protocol can be well adopted and prescribed safely to healthy adults, and possibly for special population cohorts such as the diseased and elderly, albeit more studies are required to confirm the latter. 

## 6. Strengths and Limitations of the Study

The present dispersed WAnT exercise training protocol is specifically designed around a typical office worker’s schedule and helped eliminate the barrier of lack of time for exercise. The major strength of the study was that it was conducted within a ‘real-life’ office-workplace scenario and therefore the findings have much practical applications. It is envisaged that cycle ergometers can be conveniently placed around the office workstations to allow individuals to perform the current proposed short-bouts of exercise throughout the work-day. The benefits of the present dispersed WAnT exercise protocol may also entice sedentary individuals who are in a state of ‘exercise resistance’ to start exercising because of the low time commitment required [50]. There are several limitations in the present study. Firstly, the study was conducted on healthy rather than metabolically challenged individuals, such as the obese or those with pre-diabetic conditions. Secondly, it was not possible to fully control for physical activities outside of the prescribed training program and dietary intake, although participants were instructed to maintain their normal habits. Also, the study did not control for the amount and type of foods ingested in the evening prior to the overnight fast which could have influenced the morning-after blood drawn serum lipid values. Thirdly, measures of insulin resistance were not determined using the hyperinsulinemia euglycemic clamp technique, which is considered to be the gold-standard measurement of insulin sensitivity, due to its invasive procedures. Finally, the inclusion of both sexes may potentially compound the results (e.g., insulin sensitivity) as there was no control for menstrual cycle variations when testing the female participants at pre- and post-training time points [51]. 

## 7. Conclusions

The results of the present study confirmed that a dispersed WAnT exercise training protocol, with rest periods of ~4 h between bouts is a practical and effective exercise strategy to improve cardiovascular fitness and lower-limb muscular strength. However, it had no significant impact on blood lipids or insulin sensitivity. It is suggested that this training regime may have important positive implications on functional work capacity, especially to the time-constrained office-workers. Finally, considering the high compliance rate observed, further research using a dispersed WAnT exercise training protocol to try to improve cardiorespiratory fitness and metabolic health in the diseased and challenging population cohort should be explored. 

## Figures and Tables

**Figure 1 ijerph-17-04860-f001:**
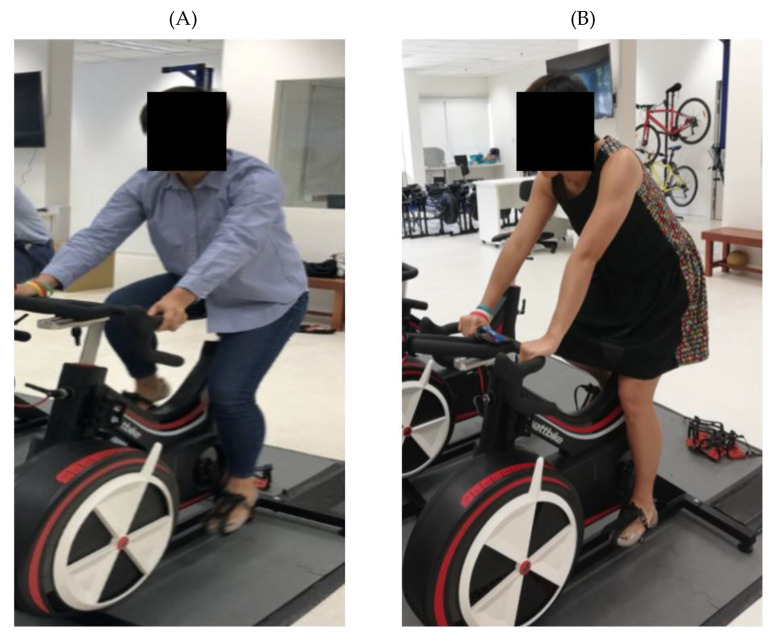
Participating individuals performing the dispersed protocol of the 30 s Wingate Anaerobic cycle Test (WAnT) bouts in two of the most common non-sporting attires; in (**A**) jeans with shirt or t-shirt, and (**B**) long-skirt.

**Figure 2 ijerph-17-04860-f002:**
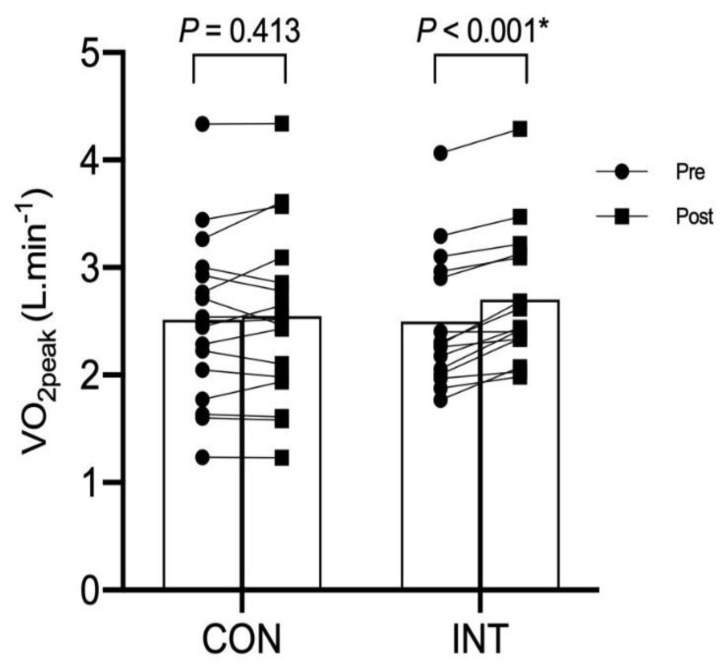
Peak oxygen uptake (or VO_2peak_) at pre- and post-6 weeks of training in the non-training or control (CON, *N* = 17) and intervention training (INT; *N* = 16) group. * Significance difference from pre-training in INT group only.

**Figure 3 ijerph-17-04860-f003:**
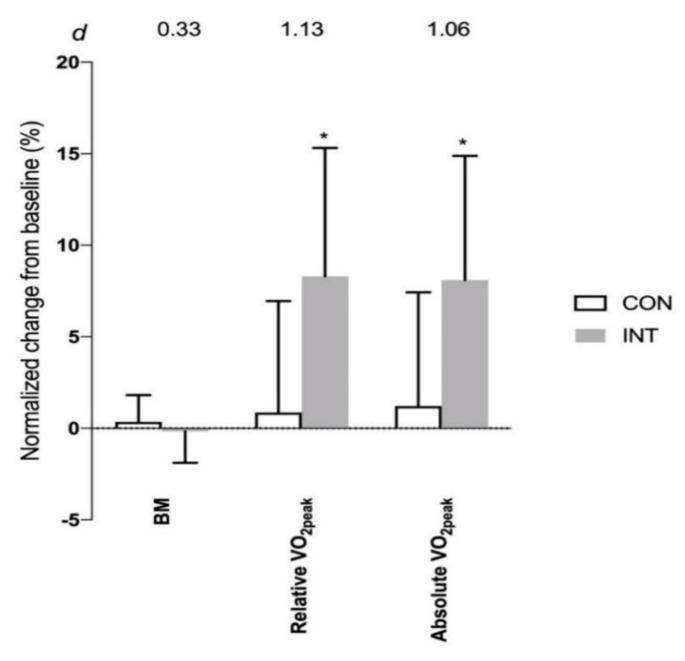
Normalized changes (mean ± SD) in body mass (BM) and VO_2peak_ from baseline (0%) to post-6 weeks training for non-training or control (CON, *N* = 17) group and intervention training (INT; *N* = 16) group. * Significance (*p* < 0.01) difference from CON group. Effect size (*d*) for the difference between CON and INT groups is shown on the top of the graph.

**Figure 4 ijerph-17-04860-f004:**
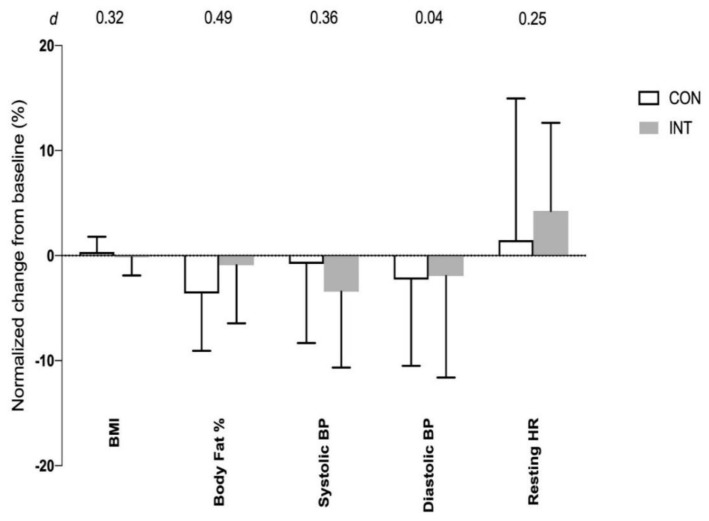
Normalized changes (mean ± SD) in body mass index (BMI), body fat percentage (%), systolic blood pressure (BP), diastolic BP and resting heart rate (HR) from baseline (0%) to post-6 weeks training for non-training or control (CON, *N* = 17) group and intervention training (INT; *N* = 16) group. Effect size (*d*) for the difference between CON and INT groups is shown on the top of the graph.

**Figure 5 ijerph-17-04860-f005:**
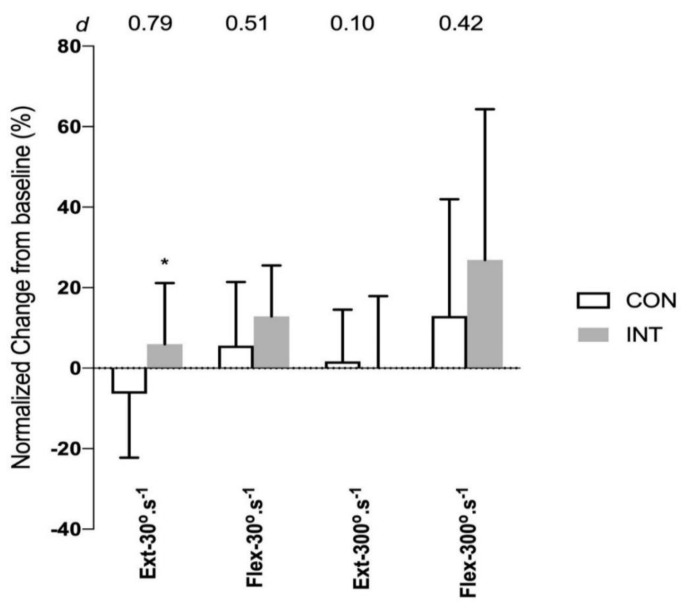
Normalized changes (mean ± SD) in maximum voluntary contraction (MVC) of the dominant lower limb on isokinetic dynamometer at knee extension at velocity of 30°·s^−1^ and 300°·s^−1^, and knee flexion at velocity of 30°·s^−1^ and 300°·s^−1^, from baseline (0%) to post-6 weeks training for non-training or control (CON, *N* = 17) group and intervention training (INT; *N* = 16) group. * Significance (*p* < 0.05) difference from CON group. Effect size (*d*) for the difference between CON and INT groups is shown on the top of the graph.

**Figure 6 ijerph-17-04860-f006:**
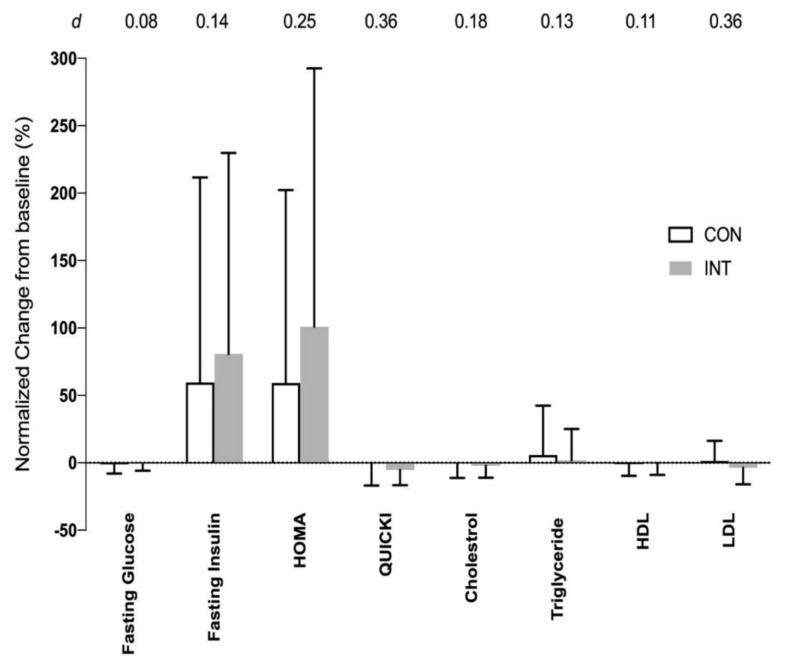
Normalized changes (mean ± SD) in fasting glucose, fasting insulin, Homeostatic model assessment (HOMA), Quantitative insulin-sensitivity check index (QUICKI), total cholesterol, triglyceride, high-density lipoprotein (HDL), low-density lipoprotein (LDL) from baseline (0%) to post-6 weeks training for non-training or control (CON, *N* = 17) group and intervention training (INT; *N* = 16) group. Effect size (*d*) for the difference between CON and INT groups is shown on the top of the graph.

**Figure 7 ijerph-17-04860-f007:**
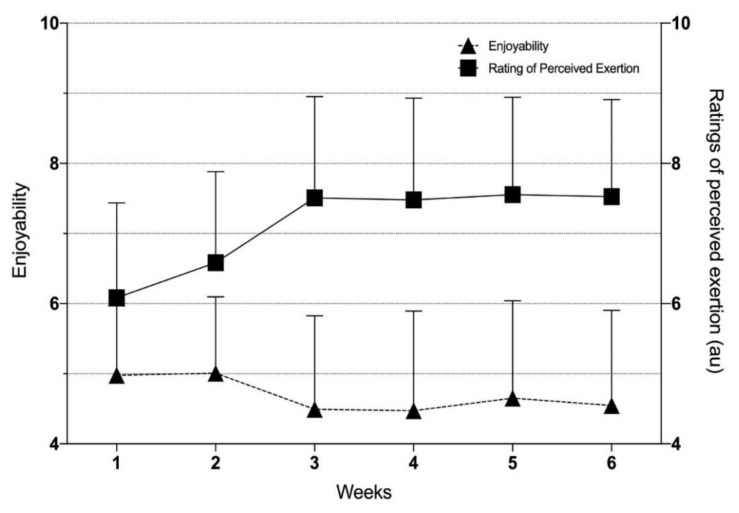
Weekly training levels of enjoyability and ratings of perceived exertion or RPE at the end of the exercise sessions across the 6 weeks of training of participants in the intervention or training group (INT, *N* = 16). The mean data is the average of all sessions performed over that week.

**Table 1 ijerph-17-04860-t001:** Comparison of time commitment to perform a clustered vs. dispersed Wingate Anaerobic test (WAnT) cycle exercise training protocol.

Exercise Component Variables	Clustered WAnT Bouts Protocol	Dispersed WAnT Bouts Protocol
Warm-up duration	1 × 60-s = 1.0 min	3 × 60-s = 3.0 min
Bouts duration	3 × 30-s = 1.5 min	3 × 30-s = 1.5 min
Recovery duration between bouts	2 × 4 min = 8 min	~4 h (but this duration is part of individual’s official work time and is hence excluded from the exercise time)
Total exercise time per day or per session	10.5 min	4.5 min
Total exercise time per week (based on 3 training days per week)	31.5 min	13.5 min (~60% less time commitment vs. clustered protocol)

**Table 2 ijerph-17-04860-t002:** Frequencies of the rest periods between the morning and afternoon exercise bouts, and between the afternoon and evening exercise bouts.

Duration of the Rest Periods	Frequencies (%)
Between 0–1 h	0
Between 1–2 h	0.88
Between 2–3 h	17.61
Between 3–4 h	31.16
Between 4–4.5 h	50.35

**Table 3 ijerph-17-04860-t003:** Baseline characteristics and measures between the two groups in the study.

Measured Variables	CON Group (*N* = 17; 8 M + 9 F)	INT Group (*N* =16; 8 M + 9 F)	*p* Value between Groups
Age (y)	35.1 ± 6.9	34.1 ± 6.3	0.68
Stature (cm)	167 ± 8	168 ± 11	0.82
Body mass (kg)	69.0 ± 14.7	69.3 ± 14.2	0.86
Body fat (%)	26.3 ± 9.8	24.9 ± 7.2	0.63
Resting Systolic BP (mmHg)	120 ± 14	117 ± 13	0.53
Resting Diastolic BP (mmHg)	79 ± 11	76 ± 8	0.33
Resting HR (b·min^−1^)	65 ± 10	63 ± 10	0.84
VO_2peak_ (L·min^−1^)	2.50 ± 0.68	2.58 ± 0.86	0.75
VO_2peak_ (mL·kg^−1^·min^−1^)	36.9 ± 9.6	36.7 ± 6.7	0.97
MVC flexion 30°·s^−1^ (Nm)	158 ± 42	168 ± 59	0.59
MVC extension 30°·s^−1^ (Nm)	91 ± 31	86 ± 25	0.66
MVC flexion 300°·s^−1^ (Nm)	80 ± 27	87 ± 31	0.46
MVC extension 300°·s^−1^ (Nm)	55 ± 23	48 ± 16	0.36
Total cholesterol (mmol·L^−1^)	4.97 ± 0.75	4.74 ± 0.93	0.45
Triglycerides (mmol·L^−1^)	2.00 ± 1.06	1.79 ± 0.99	0.56
HDL (mmol·L^−1^)	1.68 ± 0.46	1.60 ± 0.37	0.58
LDL (mmol·L^−1^)	2.87 ± 0.71	2.77 ± 0.75	0.71
Fasting Glucose (mmol·L^−1^)	4.79 ± 0.57	4.56 ± 0.42	0.21
Fasting Insulin (mmol·L^−1^)	5.96 ± 3.70	4.68 ± 3.16	0.29
Insulin resistance–HOMA (au)	1.32 ± 0.89	0.96 ± 0.71	0.22
Insulin sensitivity–QUICKI (au)	0.39 ± 0.07	0.39 ± 0.06	0.99

Key: M = males; F = females; CON = control or non-training; INT = Intervention or training; BP = blood pressure; HR = heart rate; VO_2peak_ = peak oxygen uptake; MVC = maximal voluntary contraction; HDL = High-density lipoprotein; LDL = low-density lipoprotein; HOMA = Homeostatic model assessment; QUICKI = Quantitative insulin-sensitivity check index; au = arbitrary unit.

**Table 4 ijerph-17-04860-t004:** The average peak and mean power of the morning, afternoon and evening 30-s Wingate Anaerobic cycle test (WAnT) bouts performed throughout the six-weeks training period.

WAnT Bouts of the Day	Peak Power (W)	Mean Power (W)
**Morning Bout**	838 ± 312	555 ± 189
**Afternoon Bout**	839 ± 318	552 ± 191
**Evening Bout**	850 ± 317	555 ± 194

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
