# Peer review of "Efficacy of a Six-Week Dispersed Wingate-Cycle Training Protocol on Peak Aerobic Power, Leg Strength, Insulin Sensitivity, Blood Lipids and Quality of Life in Healthy Adults"

_ijerph, 2020, doi:10.3390/ijerph17134860_

Round 1
Reviewer 1 Report
1.- In the abstract it is necessary to comment and highlight what type of physical activities the individuals studied carried out, so that the reader has an idea of ​​the type of exercise that was carried out.
2.- in the line 39. ”In line with these benefits, the World Health Organization 39
(WHO) recommends that adults engage in a minimum 150 min per week of moderate-40
intensity or 75 min per week of vigorous-intensity continuous aerobic physical activity to 41
achieve meaningful improvements in cardiorespiratory fitness and metabolic health [2] ”. We must take into account people with chronic degenerative diseases such as heart conditions and how this type of training affects their condition.
3.-on line 67, “Indeed, as little as six exercise sessions of 4 to 7 bouts of 67
30-s ‘all-out’ sprints with 4 min recovery performed over two weeks, positively impacted 68
skeletal muscle oxidative capacity [12]. ” First define what the oxidative capacity of the muscles is.
4.- In the introduction, it is necessary to add a paragraph referring to adequate nutrition, which is also a key factor in maintaining people's well-being and the regulation of different critical levels in the body.
5.- In the lines 302 and 304, there are some marks or joints in the text, which seem to have not been well reviewed before sending it. Review throughout the document.
6.- the novelty of the manuscript is poor.
7. The presentation of results is limited since there is no graph, figure or illustration of the results that can envelop the reader and give the paper an adequate presentation.
The focus and importance of the manuscript should be rethought, in addition to correcting different areas in i
Author Response
REPLY to REVIEWER 1
We would like to thank the Reviewer for his/her review feedback and comments. Please inform us if there any additional corrections and changes required.
1.- In the abstract it is necessary to comment and highlight what type of physical activities the individuals studied carried out, so that the reader has an idea of ​​the type of exercise that was carried out.
Reply: We have highlighted this aspect. Please see Line 18 in the revised manuscript. The 30-s Wingate Anaerobic cycle test (or WAnT) bout exercise is a well-known and common exercise protocol used by scientific researchers.
2.- in the line 39. ”In line with these benefits, the World Health Organization 39
(WHO) recommends that adults engage in a minimum 150 min per week of moderate-intensity or 75 min per week of vigorous-intensity continuous aerobic physical activity to achieve meaningful improvements in cardiorespiratory fitness and metabolic health [2] ”. We must take into account people with chronic degenerative diseases such as heart conditions and how this type of training affects their condition.
Reply: We have noted the Reviewer’s concern. We however would like to kindly highlight that the focus of this study was on healthy adults and that the advice or recommendation were specifically meant for the healthy population.
3.-on line 67, “Indeed, as little as six exercise sessions of 4 to 7 bouts of 30-s ‘all-out’ sprints with 4 min recovery performed over two weeks, positively impacted skeletal muscle oxidative capacity [12]. ” First define what the oxidative capacity of the muscles is.
Reply: Thank you for your comments. We have now included a sentence to define what we meant. Please see Line 73 in the revised manuscript.
4.- In the introduction, it is necessary to add a paragraph referring to adequate nutrition, which is also a key factor in maintaining people's well-being and the regulation of different critical levels in the body.
Reply: Thank you for your comment. Although the focus of this study was on exercise, we do agree with the Reviewer and have included a sentence to convey the same message. Please see Line 40 in the revised manuscript.
5.- In the lines 302 and 304, there are some marks or joints in the text, which seem to have not been well reviewed before sending it. Review throughout the document.
Reply: We have made the changes accordingly. Please see Line 342-345 and throughout the revised manuscript.
6.- the novelty of the manuscript is poor.
Reply: We are disappointed that the Reviewer has had felt this way and would like to take some time to explain our study protocol. We are pleased to have found that our clustered cycle sprint exercise protocol was 2.5x more time-efficient than a traditional or clustered sprint cycling exercise protocol, yet achieved similar cardiorespiratory fitness improvement following 6 weeks of training. This was made possible by performing a single short sprint bout at several strategic time-points of the day. We designed this exercise protocol conveniently around the participant’s daily work schedule where he or she only needed to commit 3 x 30-s of cycling (1) once before starting work (2) once before going for lunch (3) and once before going home. This will ensure that ‘time’ was NOT the main barrier for the lack of exercise. Further this study was conducted with actual office-workers in a real-world ecological valid setting – and the study’s findings has provided many practical implications.
7. The presentation of results is limited since there is no graph, figure or illustration of the results that can envelop the reader and give the paper an adequate presentation.
The focus and importance of the manuscript should be rethought, in addition to correcting different areas in i
Reply: We were unclear how the manuscript copy that was send to the Reviewer were missing all the necessary tables and figures (this omission has to be highlighted to the journal’s administrative editor who should ensure that all manuscript send out for review should be complete!). Nonetheless, the authors apologies for any inconvenience caused to the Reviewer. The authors hope that in this revised version of the manuscript, the Reviewer has all the complete main text, references, tables, and figures to have a close re-look at the manuscript proper. Please note the manuscript has 4 Tables, 7 figures and 1 supplementary table.

Reviewer 2 Report
This paper reports a test of the effects on health-related variables of a HIIT protocol spread through the working day. The paper is well written and the research well conducted. I have only minor issues that should be resolved.
The authors have chosen to present the data as change scores in figures rather than provide the actual values. At the very least, the authors should provide the pre and post values as supplementary materials. Any reader seeking to summarize effects of HIIT in the future would need the raw values.
There are two potential limitations that have not been acknowledged in the manuscript.
From the methods, there does not appear to have been any screening for levels of physical activity prior to the study. Many similar studies would restrict the sample to sedentary individuals. Based on the VO2max scores, this appears to be a relatively fit sample, with a healthy lipid profile. The authors reason that the absence of effects on lipids may reflect the relatively low training volume used in the study (line 426 ff). An alternative explanation is that the tested a healthy sample and were unlikely to improve a healthy profile.
Although participants fasted for at least 8 hours before blood sampling, there was no control for the last meal before the fast. As a result, serum lipid values may have been affected. The authors should acknowledge this as a limitation.
As there are no significant differences reported in figure 4, I am unsure of the value of the effect sizes at the top of the figure.
For all F ratios, please report the degrees of freedom.
Line 260: missing ‘not’ before ‘to engage’.
Line 295: correct the ‘P>’ to ‘P<’
Author Response
REPLY to REVIEWER 2
We would like to thank the Reviewer for his/her review feedback and comments.
- This paper reports a test of the effects on health-related variables of a HIIT protocol spread through the working day. The paper is well written and the research well conducted. I have only minor issues that should be resolved.
Reply: We would like to thank the Reviewer for his/her positive and encouraging comments.
- The authors have chosen to present the data as change scores in figures rather than provide the actual values. At the very least, the authors should provide the pre and post values as supplementary materials. Any reader seeking to summarize effects of HIIT in the future would need the raw values.
Reply: We agree with Reviewer. We have now included the pre- and post-intervention data as a Supplementary table.
There are two potential limitations that have not been acknowledged in manuscript.
- From the methods, there does not appear to have been any screening for levels of physical activity prior to the study. Many similar studies would restrict the sample to sedentary individuals. Based on the VO2max scores, this appears to be a relatively fit sample, with a healthy lipid profile.
Reply: We agree with Reviewer. We have now included the following sentences to convey the same message, Line 542-543 in the revised manuscript.
- The authors reason that the absence of effects on lipids may reflect the relatively low training volume used in the study (line 426 ff). An alternative explanation is that the tested a healthy sample and were unlikely to improve a healthy profile.
Reply: We agree with Reviewer. We have now included the following sentences to convey the same message, Line 470 to 473 and Line 504 to 507 in the manuscript.
- Although participants fasted for at least 8 hours before blood sampling, there was no control for the last meal before the fast. As a result, serum lipid values may have been affected. The authors should acknowledge this as a limitation.
Reply: We agree with Reviewer. We have now included the following sentences under our limitation section, Line 546 to 548 in the revised manuscript.
- As there are no significant differences reported in figure 4, I am unsure of the value of the effect sizes at the top of the figure.
Reply: Because we have decided to include effects sizes in our analysis of all of the results, we felt compelled to do the same for all of our analyses, no matter the significance level as a form of standardization of reporting for the entire manuscript.
- For all F ratios, please report the degrees of freedom.
Reply: We have made the changes throughout the Results section of the manuscript.
- Line 260: missing ‘not’ before ‘to engage’.
Reply: We have made the change accordingly. Line 268 in the revised manuscript.
- Line 295: correct the ‘P>’ to ‘P<’
Reply: We have made the change accordingly. Line 314 in the revised manuscript.

Round 2
Reviewer 1 Report
The content and some aspects that were recommended were improvedAuthor Response
The authors would like to thank the Reviewer for his/her comments.
Reviewer 2 Report
I have reworded the sentence starting at line 542 to clarify its meaning.
‘Firstly, the study was conducted on healthy rather than metabolically challenged individuals, such as the obese or those with pre-diabetic conditions.’
I have tweaked the sentence starting on line 546
‘Also, the study did not control for the amount and type of foods ingested in the evening prior to the overnight fast which could have influenced the morning-after blood drawn serum lipid values.’
Author Response
I have reworded the sentence starting at line 542 to clarify its meaning.
‘Firstly, the study was conducted on healthy rather than metabolically challenged individuals, such as the obese or those with pre-diabetic conditions.’
REPLY: Changed according to suggestion by Reviewer; Line 542 in the second revised manuscript.
I have tweaked the sentence starting on line 546
‘Also, the study did not control for the amount and type of foods ingested in the evening prior to the overnight fast which could have influenced the morning-after blood drawn serum lipid values.’
REPLY: Changed according to suggestion by Reviewer; Line 546 in the second revised manuscript.